

# An inventory of Arctic Ocean data in the World Ocean Database

Melissa M. Zweng[1], Tim P. Boyer[1], Olga K. Baranova[1], James R. Reagan[1,2], Dan Seidov[1], Igor V. Smolyar[1]

[1]NOAA National Centers for Environmental Information, Silver Spring, Maryland, USA
[2]Earth System Science Interdisciplinary Center/Cooperative Institute for Climate and Satellites – Maryland, University of Maryland, College Park, MD, USA

Correspondence to: Melissa M. Zweng (melissa.zweng@noaa.gov)

**Abstract.** The World Ocean Database (WOD) contains over 1.3 million oceanographic casts collected in the Arctic Ocean basin and its surrounding marginal seas. The data come from many submitters and countries, and were collected using a variety of instruments and platforms. These data, along with the derived products World Ocean Atlas (WOA) and the Arctic Regional Climatologies, are uniquely useful-- the data are presented in a standardized, easy to use format  and include  metadata and quality control information. Collecting data in the Arctic Ocean is challenging, and coverage in space and time ranges from excellent to nearly non-existent.   WOD has compiled the most complete collection of Arctic Ocean profile data, ideal for oceanographic, environmental and climatic analyses (https://doi.org/10.7289/V54Q7S16).

## 1 Introduction

The Arctic Ocean has a great influence on the earth's climate (Aagaard and Carmack, 1994) and supports vast and diverse ecosystems. Change in this region has been swift as the Arctic has warmed at a much faster rate than the lower latitudes (Serreze and Barry, 2011). Understanding the dynamics of the Arctic Ocean is critical and research requires data from both the present and the past. Unfortunately, the area is remote and the waters dangerous.  The basin is largely ice-covered in winter, rendering ships unable to collect samples at this time of year. This leads to a marked seasonal bias in the data, which lean heavily towards observations in the summer, when there is less ice cover. However, even in the summer months the seas are full of drifting ice, which can present a potential hazard to ships and other sampling platforms. Because Arctic data is difficult to gather, it is scarce and scientifically valuable.



The World Ocean Database (WOD), the largest publicly available, quality-controlled global ocean profile database, offers a wealth of Arctic Ocean ocean profile data, over 1.3 million casts of data, collected from 1849 to the present. The following will discuss the history of Arctic Ocean observations, as well as show that WOD represents a collection of data gathered throughout this history, suitable for temporal and spatial analysis of this climatically critical region.

**2 History of Arctic data**

Local populations have widely traveled and fished Arctic basin coastal areas since antiquity. The early 1800s mark the first recorded expeditions to the High Arctic and North Pole, when William Edward Parry's expedition traveled to approximately 82ºN.  Many other explorers attempted to reach the pole with varying degrees of success.  Robert Peary's 1909 expedition, achieved by ship, dogsled and on foot, was the first with a believable claim to have reached the pole.  The first surface ship

to reach the pole was the Soviet nuclear icebreaker Arktika in 1977.

Many of the polar explorers collected oceanographic data during their travels, typically by bucket or bottle samples and by recording meteorological information.  In fact, it was the scientist-explorers such as Nansen, Sverdrup and Ekman who observed the ocean as they explored during the late 1800s and early 1900s that formulated and enriched the young science of

oceanography.  The oceanic and atmospheric data they collected were published in the expedition's cruise reports.  These reports remain in library collections and many have been digitized for modern study. Rudels (2013) offers an overview of these early expeditions.

Scientists have continued to conduct research and collect data in the Arctic Ocean.  There have been concerted research

efforts for each of the International Polar Years (IPY; 1882-1883, 1932-1933, 2007-2008), as well as for the International Geophysical Year (IGY; 1957-1958). These efforts were designed to produce quasi-synoptic snapshots of the environment.

Sampling in the winter was especially difficult due to harsh weather and ice, but scientists found ways to access the winter Arctic. Drifting ice camps and buoys, research cruises by submarine, and sampling by plane and helicopter have been used to

collect data. In 1937, the Soviet Union established the first drifting ice camp, North Pole 1, using aircraft to drop researchers at the North Pole.  The camp drifted southward and eventually, after nearly a year, currents advected it out of Fram Strait and the crew was retrieved (Shirshov and Federov, 1938). Several more Soviet drifting ice camps at or near the North Pole followed (Treshnikov, 1977). The US and Canada established a number of drifting ice observational programs to study ocean and ice dynamics in the 1950's (Sater, 1964).

Another novel method to gather data in this challenging region involves instrumented marine mammals, who are natural profilers as they dive to feed and surface to breathe. The MEOP project (Marine Mammals Exploring the Oceans Pole to



Pole, http://meop.net) maintains a global database of instrumented marine mammal data. Many CTD profiles from instrumented pinnipeds in the Arctic are already available in the WOD (Fedak, 2013). Other instrumented marine mammals such as white whales (Lydersen et al., 2002) and narwhals (Laidre et al., 2010) are not available in WOD13, but have been added to the database for later releases. These animal-based approaches have the benefit of providing data during seasons

when the ocean is largely ice-covered; the marine mammals surface in the open water of leads and polynyas-- biologically vital areas-- to breathe and transmit data.

Lee et al. (2009) documents a variety of new, autonomous instrumental approaches that enhance the Arctic observing network. Ice-tethered profiling floats have been developed that are inserted in floating ice floes and profile under the ice

whether it is drifting or held fast, sampling areas that were previously inaccessible (Kikuchi et al., 2002; Toole et al., 2011; Krishfield et al., 2008). Argo floats, which have found limited use so far in the Arctic Ocean, are being modified by scientists at Universite Laval in Quebec to profile in icy regions and monitor the ice edge in Baffin Bay and the Labrador Sea (Le Traon et al., 2012). Argo floats have been deployed under seasonal ice in the Antarctic (Wong and Riser, 2011) with ice detection technology with plans to deploy in the Arctic as well.

**3 Description of Arctic Ocean and basic hydrography**

The Arctic Ocean is the smallest of the ocean basins, covering about 14 million km$^2$ (the global ocean covers 361 million km$^2$). It is an estuarine-type basin, with inflow and outflow limited to the following regions: the Bering Strait, between Alaska and Kamchatka; Fram Strait, between Greenland and Svalbard; the Barents Sea; and the straits of the Canadian Archipelago. Figure 1 shows the boundaries, basins and straits of the Arctic Ocean basin, which includes both the "high

Arctic" and its marginal seas, the Chukchi, Beaufort, Lincoln, Greenland, Norwegian, Barents, Kara, Laptev, and East Siberian seas.

The single interchange between the Arctic Ocean and the Pacific Ocean is the narrow and shallow Bering Strait. The flux through the strait is estimated to vary seasonally between 0.4 Sv (Sverdrup; 1 Sv = $1x10^6$ m$^3$ s$^{-1}$) and 1.3 Sv from the Pacific

into the Arctic (Woodgate and Aagard, 2005). The interchange between the Arctic Ocean and the Atlantic Ocean is much larger and more complex. The principal interchange, a two-way flow, occurs east of Greenland, through Fram Strait and the Barents Sea. The interchange through Fram Strait is estimated to be about 7 Sv into the Arctic Ocean basin, and 9 Sv out of the basin (Fahrbach et al., 2001), and 1-3 Sv through the Barents Sea (Schauer et al., 2002). An outflow estimated at about 1-2 Sv occurs through the Canadian Archipelago (Melling et al., 2008).

Freshwater inputs to the basin include rivers (about 0.1 Sv) and precipitation minus evaporation (P – E, about 0.06 Sv). Though a small volume, riverine input contributes a large amount of freshwater to the system (Serreze et al., 2006).



Freshwater output occurs through liquid water and drifting ice. Variations in this output may account for large salinity variations in the Atlantic Ocean such as the Great Salinity Anomaly documented by Dickson et al. (1988) (Aagaard and Carmack, 1989).

## 4 Data discovery, access and archival

The scarcity and high cost of obtaining Arctic Ocean measurements makes the data that exist particularly valuable. One challenge is convincing individuals, industry and governments that the data should be shared freely. Data gathered by military programs often remain classified for long periods of time. In particular, there were extensive Soviet and Russian surveys in the Arctic Ocean-- the most extensive long-term observing program-- that remain unavailable, except as statistical derivative products as used by, for example, Swift et al. 2005. However, there are exceptions that show cooperation between

the military and civilian scientists. The US Navy-led SCICEX ("Scientific Ice Expeditions") program provided several nuclear submarines so that civilian scientists could collect measurements under the ice. The data and results of this expedition were made public. This provided invaluable observations in a previously unstudied environment through a platform uniquely mobile under the ice (Rothrock et al., 1999). The data allowed for breakthrough analysis of the region (e.g. Morison et al., 1996).

With the recent boom in Arctic Ocean natural resource exploration, private industry collects an increasing amount of data in the area. Like data collected by the military, industry data provides an advantage to the company that has it, and it is no surprise that historically they have been reluctant to share it. However, the Deepwater Horizon oil spill in the Gulf of Mexico led to an increasing awareness of the volume of industry data holdings and an appreciation of their value to the scientific

community in understanding the environment. In particular, Shell, Statoil and ConocoPhillips signed an agreement with the US National Oceanographic and Atmospheric Administration (NOAA) to share many of their observations on the Alaska shelf in the Beaufort and Chukchi Seas. This agreement lasted from 2011 to 2016, when Shell ceased operations in the Alaskan Arctic.

Another source of data for climate study involves international cooperation with historical data archives. The International Global Ocean Data and Rescue (GODAR) project, a project of the International Ocean Data and Information Exchange (IODE), unearths data at risk of loss, in paper records and obsolete magnetic formats, and preserves them in modern digital format. WOD has worked closely with GODAR to ensure that all data rescued by the program are archived and available through WOD.

For academic scientists, there is often a reluctance to share data, in particular until scientists complete their research and publish papers based on the data. The National Science Foundation (NSF), which funds many US scientists and research,



requires principal investigators to submit a data management plan and archive their data within 2 years to ensure that the data be made accessible to the public. However, enforcement of this requirement can be difficult.

Even for publically available data, there exist logistical challenges to disseminating that data. In the case of Arctic data funded by US projects such as NSF, principal investigators may submit their data to several archives. This includes the national data centers, which are committed to the long-term archival of the data (National Centers for Environmental Information (NCEI), https://www.ncei.noaa.gov/s2n/), the Arctic Observing Network/Advanced Cooperative Arctic Data and Information Service (AON/ACADIS) archive (https://www.aoncadis.org/home.htm), now replaced by the Arctic Data Center (https://arcticdata.io), and a number of other topic- and region-specific NSF-funded data archives. Data is also collected and served as part of regional observing systems like the Arctic Ocean Observing System (AOOS, http://www.ioos.noaa.gov/regions/aoos.html). In addition, scientists outside the United States have archives to which they may be required to or prefer to submit data. At present, a user in search of data may need to visit several websites and projects and combine data in various formats.

Moving forward, it will be critical for the archives, regional associations, and data assembly centers to cooperate, providing means for users to access and collate data from multiple sources. Data archives are becoming more sophisticated in the way they serve data, allowing for activities like federated search and networked catalogs that enable people in search of data to view, access and download data from several archives at once. Projects such as DataOne (Strasser et al., 2012) are helping to familiarize scientists in the field with the standard formats that these data services require, facilitate data discovery and delivery, and work with archives to steward data for the long-term.

In this context, WOD serves a unique role: an aggregator for ocean profile data, which presents the casts in a uniform data and metadata format and provides additional services like quality control and value-added derived products. This allows the data to be used for scientific analysis without the burden of format processing, helps to unify a fractured data system and provides user-friendly access to Arctic data. As data service technology advances, features like standardized formats and web services will allow access to data from a variety of sources in a unified way. This unified system will be applied to recent and future observation systems. The WOD will continue to be the main source of historic Arctic data in this unified system.

## 5 Arctic data in the World Ocean Database

The World Ocean Database (WOD) is the largest publicly available, quality-controlled global ocean profile database, made up of data archived at the US National Centers for Environmental Information (NCEI), part of NOAA. As of this writing, WOD contains over 14.7 million casts sampled all over the world ocean. The casts are quality-controlled and converted to a

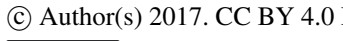

single common format for ease of use. Releases of WOD are available online at http://www.nodc.noaa.gov/OC5/WOD/pr_wod.html. The online database is updated every 3 months, and every 4 years full quality control is performed on the data and the full WOD product is released. This analysis will use the latest version of WOD, released in September 2013 (WOD13; Boyer et al., 2016).

It is a challenge to combine data from many instruments and observing platforms collected over three centuries into a coherent, convenient, and broadly useful data product. WOD data are converted to a standard format for ease of use, and metadata about the original source of the data is maintained with each cast. Rigorous automatic and manual quality control checks are performed on each cast in WOD. Quality control flags are distributed with every measurement in the cast as well

as on a cast-wide and cruise-wide level, communicating to the data user if the data failed any quality control checks. This allows users to be informed consumers and use the data of quality that fits their needs. The quality control procedures are documented in detail in Johnson et al., 2013. The instrumental precision of each instrument type included in WOD is also documented in Boyer et al., 2013.

The casts in WOD derive from oceanographic profile data in the NCEI archive.  About 10% of the casts in WOD come from the Arctic Ocean (1,389,689 casts). Figure 1 shows the bounds of the Arctic Ocean as defined in WOD.

The following sections focus on the data included in WOD13. Following the release of WOD13, the database has continued to evolve, adding new ocean profile data archived at NCEI as well as older, historical data either newly archived or

previously unprocessed data from the archive. As of March 2017, nearly 75,500 more casts have been added to WOD from the Arctic Ocean. We encourage the community to submit new data and alert us to gaps in the data and to previously unknown datasets, so we can keep WOD as up-to-date and comprehensive as possible.

## 5.1 Data distributions by time, space and instrument

Figure 2 illustrates the data density for the complete dataset.  Warmer colors have more data per 1°x1° grid square as

indicated by the legend, and areas in white have no data. The best coverage in the Arctic Ocean, from 1849 to present, is in the Eurasian sector, a slice from Iceland to Novaya Zemliya and north to about 80°N. This area is geographically well-sampled, with over 100 data points per 1°x1° grid square, and closer to the coasts with over 500 data points per grid square. 50 or more data points per 1°x1° grid square exist along western coast of Greenland and Davis Strait, northern coast of Alaska, and a swath east of Inuvik, Canada along the Beaufort Shelf edge to about 80°N.  The Russian shelf has dense

sampling, especially near river mouths and ports.

Outside of these regions, sampling falls off dramatically. Figure 2 illustrates sparse data coverage for almost all of the northernmost part of the Arctic Ocean (80°N-90°N). The small straits of the Canadian Archipelago also have poor coverage,



as does most of the East Siberian Shelf. In these regions, there may be as few as 1 to 5 samples per 1°x1° grid box over the entire 1849 – present period. In a few grid squares, particularly north of Greenland and north of the Canadian Archipelago, there are no data.

Figure 3 shows the distribution of Arctic Ocean casts through time. The Herald collected the first Arctic Ocean data found in WOD in 1849 while on a mission to rescue the Franklin expedition; the ship's logs were digitized from printed records under the GODAR project. As years passed, the amount of data collected per year gradually increased, falling off somewhat due to World War II, and then increased sharply and decreased after about 1990. The data density plot shows a decline in casts in the Barents Sea, possibly due to the fall of the Soviet Union, the most active surveyor in this area. Peak periods include
1959 (due to the IGY) and the 1980s. More than three-quarters (77%) of the data was collected between 1950 and 1990.

Mapping the data distribution per decade provides a look at the difficulty of performing basin-scale analysis over time. Figure 4 shows the data density per decade, beginning in 1900-1910. Until 1950-1960, areas outside the Eurasian shelf had very little, if any, data. However, likely due to the IGY in 1957-1958, sampling improved and data coverage increased from
that point on. After 1950, data were collected in Baffin Bay and on the Canadian/Alaskan and Russian shelves. The 1980s and later also find more data north of 80°N, much of it collected by drifting buoys and icebreaker surveys.

The largest portion of Arctic Ocean data in WOD, by number of casts, comes from moored buoys (MRB), making up just under half. However, this counting of data is somewhat misleading. Each timestep sampled counts as a cast and moored
buoys can sample up to once per minute. This leads to a large number of 'casts' for a single mooring. Bottle data (OSD) make up 27% of the casts, and the remainder are Conductivity-Temperature-Depth sensors (CTD, 7%), Mechanical Bathythermograph (MBT, 6.5%), Expendable Bathythermograph (XBT, 2%), Drifting Buoys (DRB, 5.8%), Profiling Floats (PFL, 1%), Glider data (GLD, 0.3%), and Instrumented Marine Mammal data (APB, 1.5%). Table 1 illustrates the number of casts for each data type. Figure 5 shows the distribution of samples of the different data types.

As previously mentioned, a small number of moored buoys collected a large number of casts. The observations collected by these buoys span long time periods, different in character from the snapshot nature of oceanographic casts. WOD contains data from two sets of moored buoys in the Arctic Ocean. The first is located in the Chukchi Sea, and sampled at five moorings from 2003 to 2006 as part of the research project Circulation of the North Central Chukchi Sea led by Tom
Weingartner of the University of Alaska, Fairbanks (Weingartner et al., 2005). The second is from six moorings across Nares Strait, north of Baffin Bay, from 2003 to 2010 submitted by Andreas Muenchow, University of Delaware as part of the Arctic-Subarctic Ocean Fluxes (ASOF) program (Rabe et al., 2010). The moorings collected current information using Acoustic Doppler Current Profilers (ADCPs) as well as hydrographic data with moored CTDs. However, WOD contains only the hydrographic data. The associated ADCP data is available from the NCEI archive. The data in WOD is certainly not



exhaustive. There are mooring data in the archive that has not yet been added to WOD (such as the 1997-1990 moorings across Davis Strait), and additional data held in other archives, such as the deep moorings across Fram Strait maintained by the Alfred-Wegener-Institut für Polar und Meeresforschung (AWI) (Soltwedel et al. (2005)).

The drifting buoy data (DRB) from the Arctic Ocean comes from ice drifters. Measurements from drifting ice floes are constrained by the motion of the ice in which they are deployed, but still provide a unique and generally long-lived Lagrangian perspective on the state of the Arctic Ocean. Starting in the 1990s, the Japan Agency for Marine-Earth Science and Technology (JAMSTEC) developed and deployed the JAMSTEC Compact Arctic Drifter (J-CAD) drifters (Kikuchi et al., 2002). In 2006, scientists at Woods Hole Oceanographic Institution (WHOI) developed and began to deploy an Ice-

Tethered Profiler (ITP) instrument that combines an ice drifter with a profiling float (Krishfield et al., 2008, Toole et al, 2011). Both of these instruments provide data coverage in areas previously unreachable during icy seasons. The instruments continue to take and transmit measurements regardless of whether the ice in which they are deployed is drifting or held fast.

**5.2 Data submitters and projects**

Table 2 shows the distribution of data by country: 57% of the data were submitted from institutions based in the United

States, 20% from the former Soviet Union (FSU), 10% from Norway, and 3% from Japan. However, the notion of assigning a country to data can be complicated as different information can be used to determine a country of origin (i.e. data can be based on the submitting institution, collecting institution, ship, or collecting scientists).

Table 3 lists the institutions that have submitted the largest number of casts making up about 70% of the Arctic Ocean casts

in WOD. However, 141 other groups have also contributed data. (Not all casts have institution information attached to them, although WOD includes this metadata wherever available.)

The GODAR project is one of the most successful projects to rescue large volumes of oceanographic profile data from historical sources, including those from the former Soviet Union. The Arctic Ocean casts are 19% of the entire contribution

from GODAR, which is over 1.1 million casts. In the Arctic Ocean, GODAR contributed 214,764 casts of data (16% of all Arctic Ocean casts in WOD). Of these casts, 198,999 (93%) are from the former Soviet Union and Russia, representing invaluable cooperation and collaboration between the US and these countries. Figure 6 shows the distribution of casts added to WOD through the GODAR project.

The International Council for the Exploration of the Sea (ICES) is an important contributor to the Arctic Ocean data inventory in the WOD. Currently, WOD contains 50,373 bottle casts and 22,778 CTD casts from ICES in the Arctic Ocean. ICES data are generally from the Atlantic side of the Arctic, with heavy profile density in the Norwegian and Barents Seas.



While the Arctic data from ICES is geographically constrained, it is very important to scientific research. The regions of high profile density lie along the pathways of the Atlantic Water inflow into the Arctic, allowing for several studies of the inflow variability (e.g. Furevik (2001), Smolyar and Adrov (2003), Carton et al. (2011), Korablev et al., (2014), , Yashayaev and Seidov, 2015 ). Figure 7 shows the distribution of Arctic Ocean casts submitted to WOD by ICES.

The primary projects that have contributed Arctic Ocean data to WOD are the Arctic/Subarctic Ocean Fluxes (ASOF) program (635,124 moored buoy casts), the Shelf-Basin Interaction (SBI) project (44,594 casts), the North Pole Environmental Observatory (NPEO) project (14,178 casts) and the International Arctic Buoy Program (8,240 casts). These projects combined contribute about half of all Arctic Ocean data in WOD. As with institutions, not all casts have project

information associated with them, but we include that metadata where possible.

Most casts include only temperature and salinity. (See the WOD documentation (Boyer et al., 2013) for both a definition of "cast" and information about which variables are included with each instrument type.) However, the OSD, or bottle, database contains data on many additional variables such as dissolved oxygen, nutrients, pH, tracers, pigments, and

15 biological information like plankton counts and primary productivity. There are 374,524 OSD Arctic Ocean casts in WOD13. The most represented variables are temperature and salinity. Table 4 shows the number of casts that contain each variable.

WOD contains data from many sources all over the world. Of the bottle data, most are from the former Soviet Union (FSU)

and many are from research institutes such as the Knipovich Polar Research Institute of Marine Fisheries and Oceanography (PINRO), Murmansk; Direction of the Hydrometservice, Murmansk; Murmansk Marine Biological Institute of the Russian Academy of Sciences (MMBI), the Arctic and Antarctic Research Institute (AARI) in St. Petersburg, and the Russian Navy. Partnerships with Russian and FSU contributors have been, and continue to be, an invaluable source of Arctic data, and it underscores the scope of the Russian/Soviet Union exploration of the Arctic region. Many of these casts were submitted to

NCEI through GODAR and the World Data Service for Oceanography in Silver Spring, Maryland, USA.

**5.3 Derived products**

In order to more widely distribute the Arctic data in the WOD, further quality control the data, and understand the large scale structure of the Arctic Ocean and environs, NCEI has produced a number of products specific to the Arctic region. The World Ocean Atlas (WOA), climatological mean fields of temperature, salinity, oxygen, and nutrients at standard depths for

the global ocean, is created using the data in WOD. While the Arctic Ocean is included in the WOA, a finer-scale grid and additional expert scrutiny of the data were used to create an Arctic regional climatology, (http://www.nodc.noaa.gov/OC5/regional_climate/arctic). Seidov et al., 2015 describes results of a pilot study of the Arctic Ocean and adjacent seas using this regional climatology along with data distribution analysis, demonstrating the Barents and





Nordic seas are well-covered by historical observations. Overlapping with the Arctic Ocean, a Greenland-Iceland-Norwegian Seas (GINS) Regional Climatology is available at http://www.nodc.noaa.gov/OC5/regional_climate/gin-seas-climate/. Both regional climatologies have 0.1-degree resolution, in addition to the 0.25-degree and 1-degree analyses in WOA.

In addition to these products and studies, NCEI is involved in international cooperation to increase data holdings and understanding of the Arctic region. In particular the International Atlas Series (https://www.nodc.noaa.gov/OC5/indprod.html#inter) in collaboration with Russian institutes and scientists has greatly increased public data holdings in the Russian Arctic.

## 6. Conclusions

WOD contributes to Arctic Ocean, environmental and climate science by providing a "one-stop" source of ocean data in a uniform data and metadata format, with quality control applied, that makes it simple for scientists to apply the information to their research.

Analysis of the Arctic is difficult due to scarcity of data. WOD has the best spatial and temporal coverage in the Eurasian
sector, and the data there support long-term studies. On a basin scale, the sparse distribution of data in space and time make robust analyses of change difficult.

WOD is a unique product that brings together data from many different countries and institutions, and represents a great international collaboration. This is especially true in the Arctic, where data from the former Soviet Union and Russia make
up a large and important component, particularly of historic data rescued by the GODAR project.

The Arctic data in WOD supports a number of products, including regional climatologies and climatological atlases. These products play to the strengths of the dataset, and highlight NCEI's cooperation with other Arctic institutions.

## Acknowledgments

The authors would like to acknowledge the work of OCL staff in maintaining the WOD, NCEI staff in maintaining the archive, and the data providers past and present who have archived their ocean data at NCEI.



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





**Tables**

| Country | Percentage |
|---|---|
| Moored buoy (MRB) | 49% |
| Bottle (OSD) | 27% |
| Conductivity-Temperature-Depth sensor (CTD) | 7% |
| Mechanical bathythermograph (MBT) | 6% |
| Drifting buoy (DRB) | 6% |
| Expendable Bathythermograph (XBT) | 2% |
| Profiling Float (PFL) | 1% |
| Instrumented marine mammal (APB) | 1% |
| Surface measurements (SUR) | 1% |

Table 1: Percentage of casts for each instrument type.

| Country | Percentage |
|---|---|
| United States | 57% |
| Soviet Union | 20% |
| Norway | 10% |
| Japan | 3% |
| Unknown | 3% |
| Great Britain | 2% |
| Russia | 2% |
| Canada | 1% |
| Germany | 1% |
| Iceland | 1% |
| Denmark | 1% |
| Other | <1% |

Table 2: Percentage of casts by country.



| Institution | Number of casts |
|---|---|
| University of Delaware, USA | 635,203* |
| Arctic Antarctic Research Institute (AARI), St. Petersburg, Russia | 60,896 |
| University of Alaska, Fairbanks, USA | 44,487* |
| Institute of Marine Research, Bergen, Norway | 39,994 |
| Woods Hole Oceanographic Institution, USA | 33,085 |
| Murmansk Directorate of the Russian Hydrometerological Service | 28,553 |
| Hydrometerological Service of the Russian Navy | 20,422 |
| Russia Northern Directorate of Fisheries | 15,468 |
| Murmansk Marine Biological Institute of the Russian Academy of Sciences | 23,165 |
| Japan Agency for Marine-Earth Science and Technology (JAMSTEC) | 14,275 |
| University of Washington, Seattle, USA | 10,736 |

**Table 3. Institutions submitting the most casts from the Arctic Ocean to WOD, and number of casts submitted. * Indicates a moored buoy dataset with a large number of profiles.**

| Variable | Number of Casts | Variable | Number of Casts |
|---|---|---|---|
| **Temperature** | 363207 | **CFC-11** | 1488 |
| **Salinity** | 307737 | **CFC-12** | 1446 |
| **Dissolved oxygen** | 72136 | **Nitrate+Nitrite** | 1280 |
| **Phosphate** | 42613 | **CFC-113** | 1115 |
| **Silicate** | 33808 | **Total Phosphorus** | 1081 |
| **pH** | 24777 | **Dissolved Organic Carbon** | 607 |
| **Nitrate** | 21391 | **Oxygen-18** | 544 |
| **Nitrite** | 20565 | **Particulate Organic Carbon** | 535 |
| **Plankton** | 13975 | **Primary** | 226 |





|  |  | **Productivity** |  |
|---|---|---|---|
| **Alkalinity** | 12937 | **Tritium** | 139 |
| **Chlorophyll** | 3282 | **Helium** | 136 |
| **Ammonia** | 3108 | **δCarbon-14** | 45 |
| **Total CO2** | 2060 | **δCarbon-13** | 14 |
| **Phaeophytin** | 1618 | **δHe-3** | 134 |
|  |  | **TOTAL CASTS** | 374524 |

**Table 4. Number of Arctic Ocean casts in the bottle (OSD) database that contain each data variable.**





**Figures**

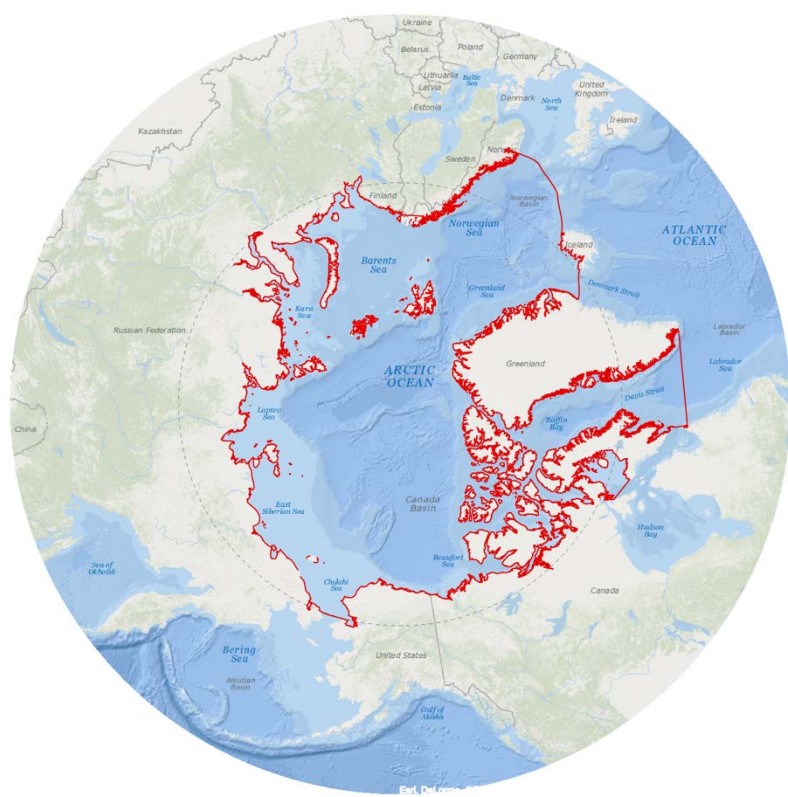

**Figure 1: Bounds of the Arctic Ocean Basin in WOD. This includes both the "High Arctic" ocean basin proper, and its marginal seas, the Chukchi, Beaufort, Lincoln, Greenland, Norwegian, Barents, Kara, Laptev, and East Siberian seas**



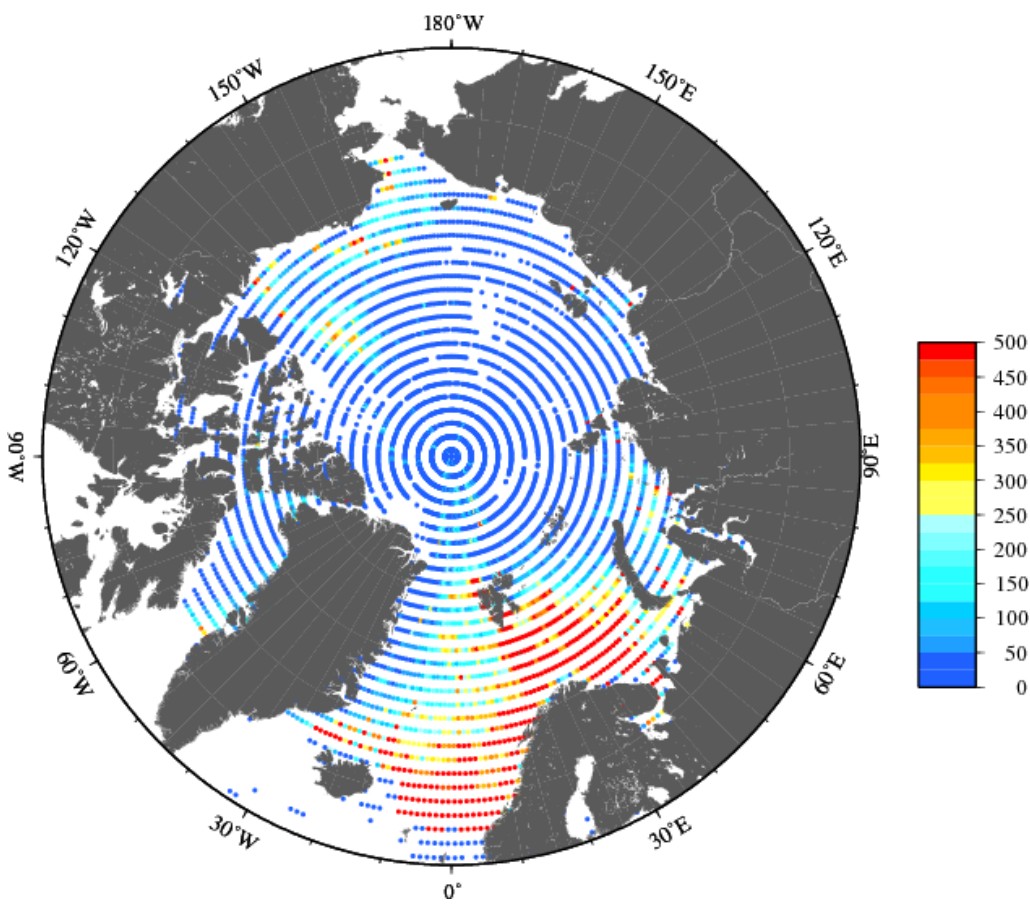

**Figure 2: Data density of Arctic Ocean casts in WOD13.**




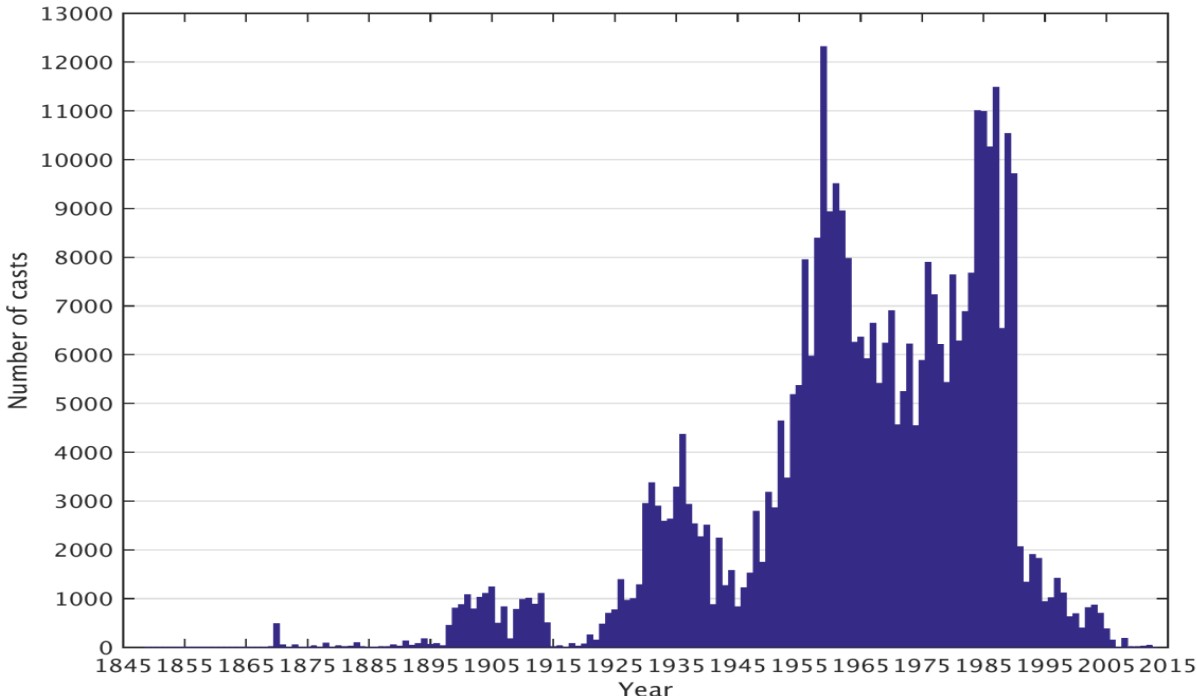

**Figure 3: Time series of number of casts per year in WOD13. Note the peaks around the International Polar Years (IPY; 1882-1883, 1932-1933, 2007-2008), as well as for the International Geophysical Year (IGY; 1957-1958).**



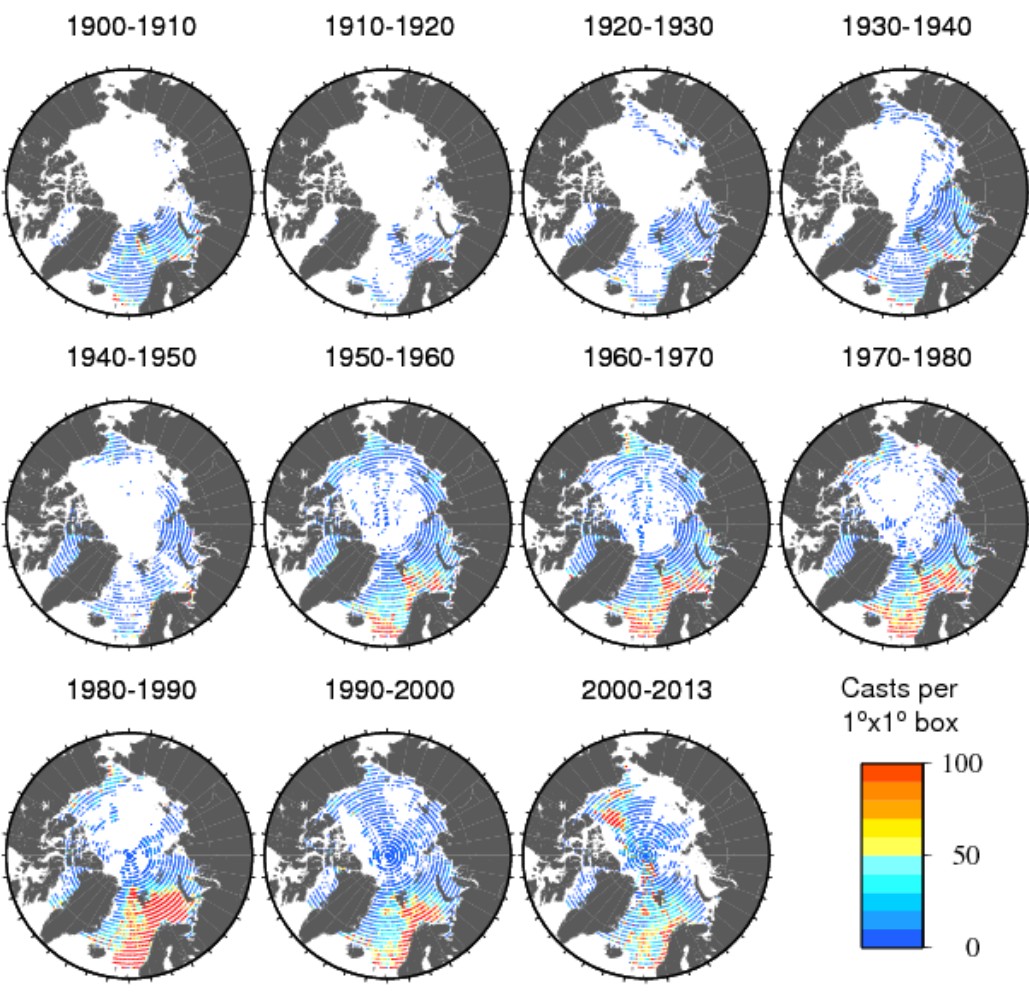

**Figure 4: Data distribution by decade.**





**Figure 5: Data distribution by instrument.**



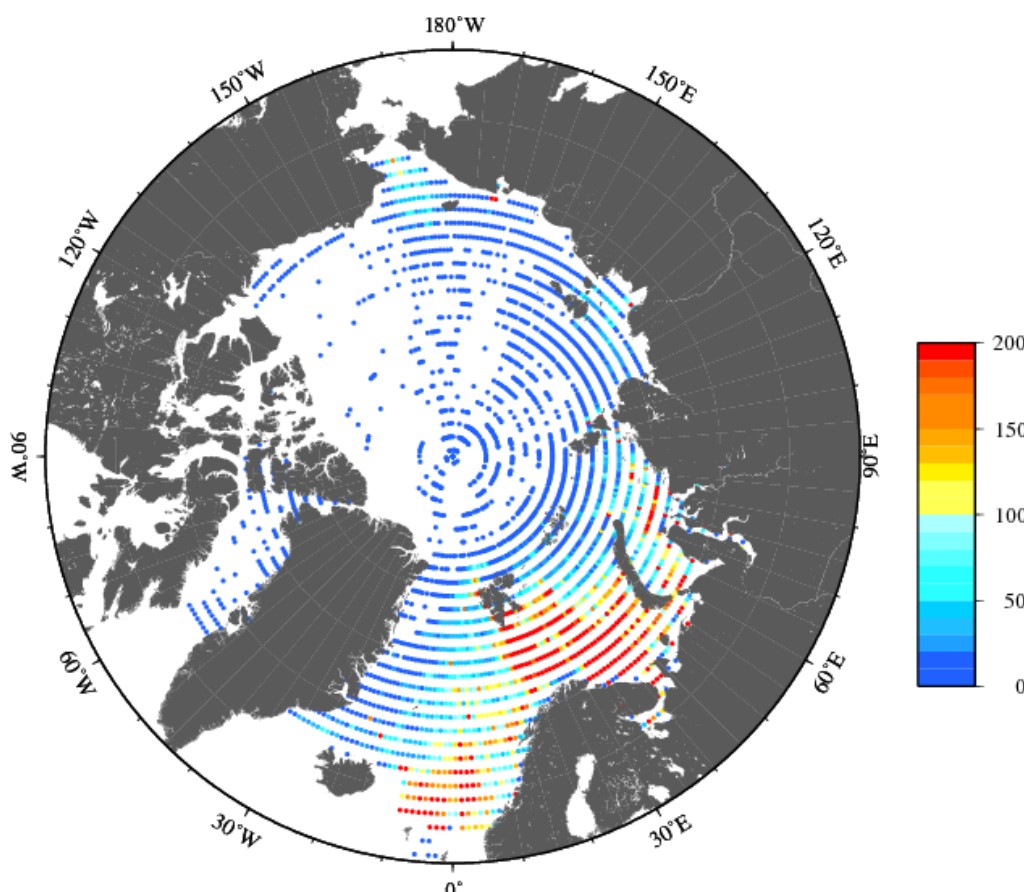

**Figure 6: Data density of casts received through the GODAR project.**

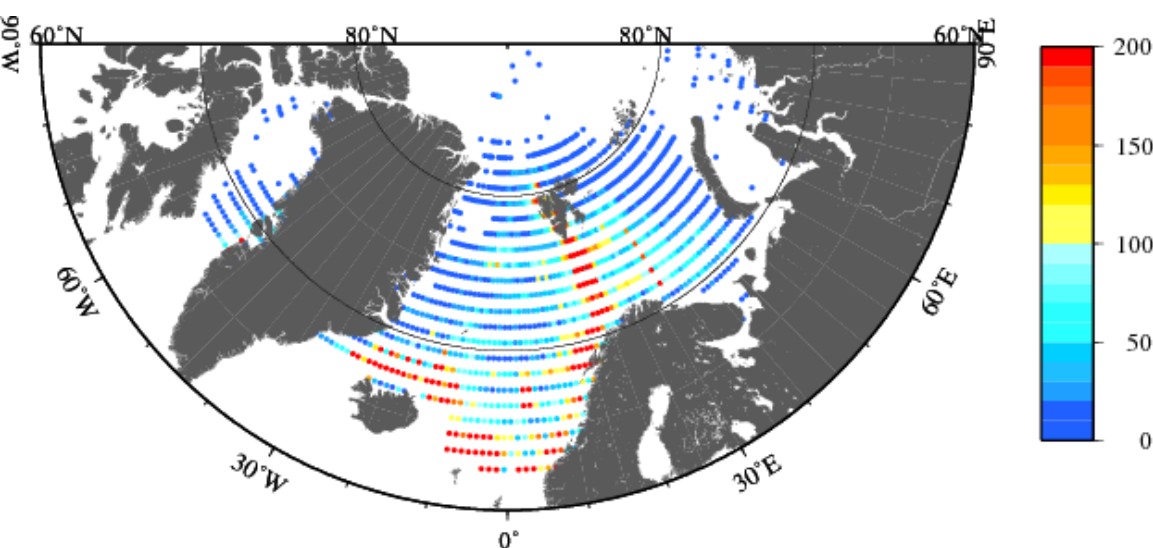

**Figure 7: Data density of casts received from ICES.**