# Peer review of "An inventory of Arctic Ocean data in the World Ocean Database"

_Earth System Science Data, 2017_

## Referee Comment (RC1) · M. Steele (Referee) · 2 Sep 2017

Review of Zweng et al., "An inventory of Arctic Ocean data in the World Ocean Database," submitted for publication to Earth System Science Data Discussions.

This is a very valuable contribution to the data science literature. I support publication after minor revision.

P1, L8: What is the definition of a "cast" (or a "profile")? For example, does a temperature profile from the surface down to 50 or 100 m depth qualify? What about down to 20 m? What about a salinity profile from 100 m depth to 150 m depth? Or a mooring with sensors at just a few depths, e.g., 50 m, 100 m, and 200 m? IE, what are the (1) upper/lower depth bounds, (2) minimum number of depth levels, (3) other thresholds

that determine a "cast?"

P2, L7: "...first recorded European expeditions..."

P3, L3: "...were not available..."

P3, L28: "...1-3 Sv into the Barents Sea from the Norwegian Sea..."

P4, L2-3: (1) "...variations in the North Atlantic Ocean..." (2) cut Aagaard and Carmack 1989 reference.

P4, L9: Swift et al (2005) and also: Steele, M. and W. Ermold, Steric sea level change in the Northern Seas, J. Climate, 20(3), 403–417, 2007.

P4, L14: Morison et al. (1996) and also: Steele, M. and T.ÂăBoyd, Retreat of the cold halocline layer in the Arctic Ocean, J. Geophys. Res., 103, 10,419-10,435, 1998.

P5, L17: What is a "federated search?" What is a "networked catalog?"

P5, L24: What is "format processing?"

P5, L30 (and P10, L14-15): It would be useful to provide some context to these statements. IE there are other global hydrographic data bases available; you should prove that WOD is the largest, as you claim. Also, is WOD the largest in the Arctic region?

P6, L16: Is this sentence "Figure 1 shows..." really necessary, given the earlier Figure 1 reference on P3, L19?

P6, L29: "...east of Inuvik..." ??? Inuvik is not on the coast. I do not see this swath.

P6, L29-30: "The Russian shelf has dense sampling..." It seems like this is really an overstatement. Most Russian arctic shelves have very poor sampling.

P7, L18-19: The very frequent sampling from MRBs really skews your statistics. You note this, which is good, but can you do better? EG can you provide a second set of statistics where you calculate daily mean profiles from each data type? Or just subsample to get once-daily sampling?

P7, L21-24: Is it necessary to include these numbers in the text, given that they are in the Table as well?

P8, L1: "…has not yet been added…" Why not, considering that these data sets started a long time ago?

P8, L5: Did you include UpTempO buoy data? Eg, https://arcticdata.io/catalog/#view/doi:10.18739/A2508R

What about Salargos buoy data? ftp://ftp.nodc.noaa.gov/nodc/archive/arc0001/9800040/1.1/data/0-data/atlas/html/dig/digargos.htm

P8, L15-18: So, which method did you use in Table 2?

P9, L6-10: Did you use Switchyard data? Eg: https://arcticdata.io/catalog/#view/doi:10.18739/A22G78

Figure 1: (1) You should mark the "High Arctic" ocean basin proper. (2) 80N is prominently discussed in the text: It should be marked here or on another figure. (3) The caption refers to the Lincoln Sea, but it is not marked on the map. (4) Your red contour includes the Iceland Sea, the Canadian Arctic Archipelago (a.k.a. the "Canadian Polar Shelf Sea"), Baffin Bay, and Davis Strait / Northern Labrador Sea, yet these are not included in your caption text. (5) Why is only 1 out of 4 deep basins in the Arctic Ocean marked? IE Canada, Makarov, Amundsen, and Nansen. Or C + M = Amerasian Basin; A + N = Eurasian Basin.

Figure 2: (1) The color scale is missing the units (just copy from Figure 4). (2) Should note that gaps = missing data (ie take this from the text and put it in the caption). (3) You should note the bias in this display of data density via lat/lon bins. IE even if data sampling was ok in the far north, the convergence of meridians means that it is likely that blue colors would still dominate. I realize that WOA is on a lat/lon grid, but that does not mean that you need to display your WOD results on this inappropriate grid for an Arctic paper. IE a better figure and statistics would use an EASE grid or similar.

Figure 3: (1) I do not see the peaks at 1882-1883 nor at 2007-2008. (2) I suggest putting the horizontal axis tick marks on the outside of the plot, so that they are visible apart from the blue bars. (3) And then you can mark these special observing periods IPY, IGY with thicker ticks or arrows.

Figure 5: (1) Can you make the lat/lon lines thicker and whiter on the land? (2) Colors are very hard to see when dots are tiny. Please make the colored dots on the key below the map larger. (3) I suggest that you note the locations of the MRB stars in the caption text, because they are very hard to find otherwise. (4) I suggest that you explain the acronyms in the caption text. (5) What is the order of overplotting here? IE which symbol is first, second, etc? But in fact, overplotting is not optimal. Another option might be to find the most common instrument in each grid cell, and color the cells that way. This method avoids overplotting and creates a much cleaner figure.

---

## Short Comment (SC1) · 20 Sep 2017

Hi Mike, thank you so much for your detailed review and thoughtful comments. I agree with your comments and will incorporate them in the revised text.

The issue of moored buoy data in WOD is an important one. You are correct in that there are many mooring datasets currently not in WOD. We've been thinking a lot about how to effectively combine these point time series data with the more geographicly dispersed profile data. A Moored Buoy Database is also in the works to put these data into a common netCDF format and make them discoverable and subset-able.

In my "kitchen sink" analysis of Arctic Data in WOD I've conflated the mooring and profile data, despite the fact that they're very different and should probably be looked

at separately.

The UpTempO and Salargos buoys are not in WOD; the Switchyard CTD data is. WOD is limited to data held at NCEI (we have archived the Salargos and Switchyard data). The NCEI archive should receive much of the NSF ADC (previously ACADIS) data in the next couple years as the long-term repository for that data center.

---

## Referee Comment (RC2) · Anonymous Referee #2 · 27 Nov 2017

P1, Abstract: Time period of the collected data should be clarified in abstract.

P1, L8: definition of 'casts' is unclear for me.

P2, L3-4: In the paper, data distributions are discussed from different perspectives. ".. WOD represents a collection of . . ." should be described more details.

P3, L3: It is better to add short description about "WOD13".

P5, L22-28: WOD provides data by three different format types (WOD native ASCII format, CSV format and netCDF format), and user selects one of them. To read WOD native data format, the user requires ODV software package or use source codes in FORTRAN, C, Matlab etc. . . . The sample codes are provided. These information should be given in the sentence.

[Figure]

P6, L18: "... in WOD13." should refer to "URL:http://data.nodc.noaa.gov/woa/WOD/DOC/wod_intro.pdf"

P7, L10: "... (due to the International Geophysical Year (IGY))"

P7, L13: Is it "Until 1940-1950, ..."?

P8, L23: "The International Global Ocean Data (GODAR) project ..."

P8, L23: When is the GODAR project period? What types of data were mainly provided by the project?

P9, L3: "... Korablev et al. (2014), Yashayaev and Seidov (2015))"

P9, 6-7: when is the period of the data covered by ASOF, SBI, NPEO and International Arctic Buoy Program?

P9, L12-17: this sentence describes about variable types. I suggest to move the sentence in 5.1 and change the subtitle as "Data distributions by time, space and instrument and variable types".

P9, L13: "... instrument type. However..."

P11, L2: Aagaard, K. and Carmack, E.C.: The role of sea ice and fresh water in the Arctic circulation, J. Geophys. Res., 94, 14, 14485-14498, 1989.

The style of other references should be followed as above.

Figure 2: 80N should be marked in the figure.

Figure 4: 80N should be marked in the figure.

Figure 5: It is better to show what the abbreviations in the figure mean .

Figure 5: Since some points are overlapped, they are unclear.

CSV data file: References of NODC code, WOD code and OCL code should be provided in the data file.

NetCDF data file: >The data format does not comply with CF convention. I suggest to use CF checker http://puma.nerc.ac.uk/cgi-bin/cf-checker.pl

>https://www.nodc.noaa.gov/OC5/WOD13/ should be added in the global metadata as "reference".

>NetCDF file should include minimum global and variable metadata for the data in the file. "keywords" is highly recommended in global metadata. For "keywords", you may choose from GCMD science keywords or something else. "Licence" is also recommended. An example of "licence" is "This data is made freely available by NODC. User must display the this citation in any publication as: < Boyer, T.P., J. I. Antonov, O. K. Baranova, C. Coleman, H. E. Garcia, A. Grodsky, D. R. Johnson, R. A. Locarnini, A. V. Mishonov, T.D. O'Brien, C.R. Paver, J.R. Reagan, D. Seidov, I. V. Smolyar, and M. M. Zweng, 2013: World Ocean Database 2013, NOAA Atlas NESDIS 72, S. Levitus, Ed., A. Mishonov, Technical Ed.; Silver Spring, MD, 209 pp., http://doi.org/10.7289/V5NZ85MT>". Variable metadata "units" is highly recommended for each variable. For instance, "salinity" variable does not have "unit" metadata.

>It is sometimes difficult to guess what variable is given from variable name. "long_name" and "standard_name" are highly recommended for all variables as variable metadata.

>If all values of a variable are same, it is not necessary to add it as a variable. For instance, all flag values for a variable 'X' take 0, I suggest to include it as "flag" metadata in variable "X".

>Data structure should be as simple as possible. For instance, the following two variables (exampleA) are able to summarize into one variable (example B). :

exampleA )

short Salinity_WODflag(z) ;

_________Salinity_WODflag:flag_definitions = "WODf" ;

short WODf ;

________WODf:long_name = "WOD_observation_flag" ;

________WODf:flag_values = 0s, 1s, 2s, 3s, 4s, 5s, 6s, 7s, 8s, 9s ;

________WODf:flag_meanings   =   "accepted   range_out   inversion   gradient anomaly   gradient+inversion   range+inversion   range+gradient   range+anomaly range+inversion+gradient" ;

Example B)

short Salinity_WODflag(z) ;

________Salinity_WODflag:long_name = "WOD_observation_flag" ;

________Salinity_WODflag:flag_values = 0s, 1s, 2s, 3s, 4s, 5s, 6s, 7s, 8s, 9s ;

________Salinity_WODflag:flag_meanings = "accepted range_out inversion gra-dient anomaly gradient+inversion range+inversion range+gradient range+anomaly range+inversion+gradient" ;

>Data structure is complex since several variables are included in one file. To make groups using netCDF4 format might be easier to understand for users.

---

## Author Comment (AC1) · 4 Jan 2018

Dear Referee #2, thank you for taking the time to review this paper. We appreciate your careful look at the manuscript, particularly the requests for clarification of terms and information, and will revise the manuscript based on your suggestions.

P1, Abstract: I will include the time period of the data.

P1, L8: The definition of "cast" is one that often causes confusion, and I was remiss in not providing more detail. A cast is a single profile taken concurrently, and the definition of a 'profile' is a set of measurements for a variable taken at discrete depths in the water column. For an instrument like a moored or drifting buoy that does not move up or down in the water column, a 'profile' is a time snapshot of the measurements

taken at different depths by the instruments on the mooring chain. I will include this information in the updated manuscript.

P2, L3-4: I sought to introduce the WOD concept but deferred the detailed explanation of it until section 5. I will add more detail to this paragraph.

P3, L3: Thank you for pointing out that I mentioned WOD13 before explaining what it was. I will correct this.

P5, L22-28: I will add detail about WOD download methods and formats here.

P6, L18: I will include a reference to the data documentation here.

P7, L10: The acronym is spelled out on P2, L21, but I will write it out again here for clarity.

P7, L13: I would say 1950-1960 is the first decade with geographically broader sampling, but the data in Baffin Bay and the Chukchi Sea in 1940-1950 do make a strong case.

P8, L23: GODAR is spelled out on P4, L26, but I will do it again here. I will also include a citation with more information about the project.

P9, L3: Thank you! I will correct this.

P9, L6-7: I will include the dates of these project data.

P9, L12-17: This is a logical suggestion and I will move the text as described.

P9, L13: The parentheses are correct here.

P11, L2: I will make changes as described.

Figure suggestions are noted and agreed with; I will make changes accordingly.

Finally, we very much appreciate your close look at the WOD netCDF format. Tim Boyer (tim.boyer@noaa.gov) maintains WOD and its netCDF format and provided a

response that addresses many of your concerns, and has made several corrections to the format based on your comments. He is also happy to discuss further questions and encourages you to contact him directly. Please see the following from Tim:

Note regarding all netCDF comments - the World Ocean Database (WOD) netCDF contiguous ragged array format has been revamped over the last six months. The new format is expected to be available by the end of January, 2018. An example file (all CTD data for 2017 - through Sep. - in WOD) is available at ftp://ftp.nodc.noaa.gov/pub/WOD/SELECT/wod_ctd_2017.nc

NetCDF data file: >The data format does not comply with CF convention. I suggest to use CF checker http://puma.nerc.ac.uk/cgi-bin/cf-checker.pl

- Thank you. Following the reviewer's recommendation and using the CF compliance checker reveals the use of type SHORT for arrays [variable]_row_size instead of type INTEGER. This has been corrected. After this correction, the format is compliant with CF contiguous ragged array conventions.

>https://www.nodc.noaa.gov/OC5/WOD13/ should be added in the global metadata as "reference".

- In the updated version of the format, the following is given as reference: references = "World Ocean Database 2013. URL:http://data.nodc.noaa.gov/woa/WOD/DOC/wod_intro.pdf

>NetCDF file should include minimum global and variable metadata for the data in the file. "keywords" is highly recommended in global metadata. For "keywords", you may choose from GCMD science keywords or something else. "Licence" is also recommended. An example of "licence" is "This data is made freely available by NODC. User must display the this citation in any publication as: < Boyer, T.P., J. I. Antonov, O. K. Baranova, C. Coleman, H. E. Garcia, A. Grodsky, D. R. Johnson, R. A. Locarnini, A. V. Mishonov, T.D. O'Brien, C.R. Paver, J.R. Reagan, D. Seidov,

I. V. Smolyar, and M. M. Zweng, 2013: World Ocean Database 2013, NOAA Atlas NESDIS 72, S. Levitus, Ed., A. Mishonov, Technical Ed.; Silver Spring, MD, 209 pp., http://doi.org/10.7289/V5NZ85MT>". Variable metadata "units" is highly recommended for each variable. For instance, "salinity" variable does not have "unit" metadata.

- Good suggestions. We have discussed 'keywords'. The problem with keywords is that not all files (especially when ragged array netCDF is implemented in our request software - WODselect) will contain each type of data. There are some files without temperature profiles for instance. We have discussed more general keywords from GCMD but at this time have decided not to include any in our global attributes. This may change in the future.

As for license, this has been another source of discussion. Since the WOD is a U.S. Government data set, it is by law available without restriction - even the restriction of requiring citation. For this reason, we have not included the license keyword.

Finally, as part of the format update, units has been added as an attribute to most variables. Some few, such as 'originators_cruise' still do not have units. But all of the measured variables do have units - with the exception of salinity which is unitless.

>It is sometimes difficult to guess what variable is given from variable name. "long_name" and "standard_name" are highly recommended for all variables as variable metadata.

- long_name and/or standard_name are now included for each variable.

>If all values of a variable are same, it is not necessary to add it as a variable. For instance, all flag values for a variable 'X' take 0, I suggest to include it as "flag" metadata in variable "X".

- The problem with doing this is that there is no way a priori to know which files will have all flag values of 0. And if there were, to have one file with flag as metadata since all values are 0 and other files with flag as a variable since there are non-zero flags
would be inconsistent and could create problems for reading software.

>Data structure should be as simple as possible. For instance, the following two variables (exampleA) are able to summarize into one variable (example B). : exampleA ) short Salinity_WODflag(z) ; _________Salinity_WODflag:flag_definitions = "WODf" ; _________Salinity_WODflag:flag_meanings = "accepted range_out inversion gradient anomaly gradient+inversion range+inversion range+gradient range+anomaly range+inversion+gradient" ;

- This has been done in the updated format.

>Data structure is complex since several variables are included in one file. To make groups using netCDF4 format might be easier to understand for users.

- We discussed using groups - and in fact we do use groups for plankton data which are much more complex than ocean profile data. The problem with groups is that many applications don't recognize them. For instance, our own THREDDS server simply ignores plankton data in WOD since it cannot interpret the group information. Many, but not all applications have moved from netCDF3 to netCDF4, but even some of those which have made the switch are not equipped to handle groups.

―――――――――――――――――――